Screening microalgae isolated from urban storm- and wastewater systems as feedstock for biofuel

Massimi Rebecca
http://orcid.org/0000-0003-2947-574X Kirkwood Andrea E. andrea.kirkwood@uoit.ca
Faculty of Science, University of Ontario Institute of Technology , Oshawa, Ontario , Canada
Valle Rogerio
Electronic publication date: 2016 Sep 1
Publication date: 2016
Volume: 4
Electronic Location ID: e2396
Received 2016 Jun 1; Accepted 2016 Aug 2
Copyright: © 2016 Massimi & Kirkwood
Copyright year: 2016
Copyright holder: Massimi & Kirkwood
License: This is an open access article distributed under the terms of the Creative Commons Attribution License, which permits unrestricted use, distribution, reproduction and adaptation in any medium and for any purpose provided that it is properly attributed. For attribution, the original author(s), title, publication source (PeerJ) and either DOI or URL of the article must be cited.
License URL: https://creativecommons.org/licenses/by/4.0/

Keywords: Algae, Fatty acids, FAME, 18S rRNA, Wastewater, Biofuel, Metal tolerance

Funding: FedDev Ontario grant BioFuelNet grant Funding for this work was provided by a FedDev Ontario grant to A. Kirkwood and a BioFuelNet grant to A. Kirkwood. The funders had no role in study design, data collection and analysis, decision to publish, or preparation of the manuscript.

==============================
Exploiting microalgae as feedstock for biofuel production is a growing field of research and application, but there remain challenges related to industrial viability and economic sustainability. A solution to the water requirements of industrial-scale production is the use of wastewater as a growth medium. Considering the variable quality and contaminant loads of wastewater, algal feedstock would need to have broad tolerance and resilience to fluctuating wastewater conditions during growth. As a first step in targeting strains for growth in wastewater, our study isolated microalgae from wastewater habitats, including urban stormwater-ponds and a municipal wastewater-treatment system, to assess growth, fatty acids and metal tolerance under standardized conditions. Stormwater ponds in particular have widely fluctuating conditions and metal loads, so microalgae from this type of environment may have desirable traits for growth in wastewater. Forty-three algal strains were isolated in total, including several strains from natural habitats. All strains, with the exception of one cyanobacterial strain, are members of the Chlorophyta, including several taxa commonly targeted for biofuel production. Isolates were identified using taxonomic and 18S rRNA sequence methods, and the fastest growing strains with ideal fatty acid profiles for biodiesel production included Scenedesmus and Desmodesmus species (Growth rate (d−1) > 1). All isolates in a small, but diverse taxonomic group of test-strains were tolerant of copper at wastewater-relevant concentrations. Overall, more than half of the isolated strains, particularly those from stormwater ponds, show promise as candidates for biofuel feedstock.

Introduction

As the earth’s human population increases, world energy demands and reliance upon fossil fuels has continued to rise. Consequently, global carbon emissions, including green-house gases, have increased and contributed to global climate warming. The need to move to fossil-fuel alternatives is not only necessary for addressing anthropogenic climate change, but also depleting world energy-stores (Höök & Tang, 2013). Algal biofuels in particular have been identified as an exceptional source of carbon neutral, renewable energy (Sharif Hossain et al., 2008; Schenk et al., 2008; Clarens et al., 2010). Their high photosynthetic efficiency, biomass production, and ability to accumulate relatively large amounts of triacylglycerides (TAG) for conversion to fatty acid methyl esters (FAME) have made them a desirable alternative to land-plant biofuel crops. Unlike first and many second-generation biofuel feedstocks, microalgae can be cultivated using saline, brackish or wastewater streams on non-arable land. Additionally, algal feedstock can be harvested batch-wise throughout the year from industrial-scale photobioreactors, reducing land, nutrient, and fresh water requirements (Guischina & Harwood, 2006; Graef et al., 2009). The success and economic viability of a microalgae-based biofuel industry will depend on several factors, including the selection of robust strains that exhibit exceptional growth rates, suitable biofuel lipid profiles, resistance to disease and predation, and tolerance to a wide range of environmental parameters (Griffiths & Harrison, 2009). Tolerance to environmental conditions is an important, but often over looked characteristic when bioprospecting for biofuels, and offers several benefits such as ease of cultivation and resiliency.

Griffiths & Harrison (2009) outlined a framework in the selection of appropriate algal strains. It identified the selection of fast-growing strains optimized for local climatic conditions as a fundamentally important quality. They also argued that lipid content as a key characteristic may not be as critical as previously thought, because fast growth encourages high biomass productivity, and high biomass density can increase yield-per-harvest volume. Research to increase growth rates and lipid production has identified optimal temperatures, pH, photoperiods and nutrients for several species (Lv et al., 2010; Sforza et al., 2012; Adams et al., 2013). These parameters, however, are dissimilar among strains and require individual optimization.

Moving forward, it is recognized that bioprospecting for biofuel microalgae will require assessments of individual and mixed-cultures of strains (Zhou et al., 2011). Additionally, ideal candidate strains may need an inherent tolerance to sub-optimal growth conditions, since wastewater from a variety of sources may be the most economically and environmentally feasible growth medium. As a first step in assessing temperate-climate microalgae for their potential as biofuel feedstock, we aimed to isolate and culture strains from several engineered wastewater systems that included urban stormwater ponds and a municipal wastewater system. These environments were targeted because they are known to support algal communities, but also experience fluctuating conditions that include variation in contaminant loads and residence-time.

For example, we know that stormwater ponds typically have elevated metal concentrations (Campbell, 1994; Marsalek & Marsalek, 1997; Karlsson et al., 2010), including the ponds in our study (Vincent & Kirkwood, 2014). It also has been well documented that municipal wastewaters have elevated levels of metals and other contaminants (Chambers et al., 1997; Gagnon & Saulnier, 2003; Wang et al., 2005; Principi et al., 2006). Hence, these engineered environments were considered promising source habitats for microalgae with inherent tolerance to fluctuating conditions and contaminant loads, including metals. Overall, the primary goal of this study was to elucidate if certain algal taxa from these wastewater source-locations possessed growth, fatty acid, and metal-tolerance traits suitable for biofuel-feedstock production.

Materials and Methods

Sample collection and strain isolation

Three 1-L water samples were collected from nine urban stormwater ponds and two natural reference ponds across Durham Region, Ontario, Canada between June and August, 2011. All of these ponds have been previously described using a similar numbering scheme in Vincent & Kirkwood (2014). In June 2011, 3-L of municipal wastewater was collected from each of three locations along the secondary treatment system of the Corbett Creek activated sludge Water Pollution Control Plant in Whitby, Ontario, Canada. Additionally, several litres of water were collected from Lake Ontario at a single nearshore location in Whitby, Ontario to expand the isolations from natural, local habitats.

Sub-samples of all collected water were transferred to sterile 250-mL Erlenmyer flasks stoppered with cotton bunts on a light bank table (12:12 dark/light cycle) in triplicate. Upon confirmation of algal growth by visual inspection, 10-mL sample aliquots were transferred to 250-mL Erlenmyer flasks and enriched with 90-mL of two defined media recipes: BG11 (Rippka et al., 1979) modified by reducing NaNO3 by one-tenth stock concentration; and CHU10 (Stein, Hellebust & Craigie, 1973). Two media types were employed to broaden isolation success for a diversity of strains, and both had nitrogen and phosphorus concentrations on par with ranges typically found in the municipal-wastewater system of our study (A. Kirkwood, 2011, unpublished data) and stormwater ponds (Vincent & Kirkwood, 2014). F2 vitamin mixture (Guillard, 1973) was added by syringe filter (Progene) to both media types after autoclaving. Isolation of unialgal strains was accomplished by methods outlined by Andersen & Kawachi (2005). In brief, mixed-culture samples were serially diluted, one in ten with sterile millipore water and spread plated on to 1.5% agar plates. Visible and distinct colonies were pegged with sterile loop and re-cultured in BG11 or CHU10 media and examined microscopically to verify they were a single strain throughout the study period, with re-culturing every three weeks. Isolated strains were observed using an EVOS XL core inverted microscope at 400× to ensure unialgal status, and subsequent sequencing of strains confirmed this. Tentative taxonomic strain identification was assigned based upon morphological descriptions in Sheath & Wehr (2003).

Growth assays

Growth assays were conducted under standardized conditions to determine growth rate and generation time for all isolates. A 10-mL inoculum of exponentially-growing stock-culture was aseptically introduced to 90-mL of sterile isolation media and grown in cotton-stoppered flasks on a cool-white fluorescent plate-glass light table (20–22 °C, 12:12 dark/light cycle) and shaken by hand once a day. Algal cell concentration was determined by microscopic enumeration using a Brightline haemocytometer on a VistaVision compound light microscope at 100× magnification. The optical density of 1-mL samples was measured at 550 nm every 24-h for a seven-day period using the Genesys 10S UV/VIS Spectrometer. These data were used to construct growth curves for each algal strain. Specific growth rate (μ), and generation time (Th) were calculated as per Guillard (1973) over a period of exponential growth.

Antibiotic treatment of algal isolates

Great care was taken to attain axenic cultures to remove the influence of bacteria on FAME profiles (Stemmler, Massimi & Kirkwood, 2016) using a modified antibiotic treatment method adapted from Jones, Rhodes & Evans (1973). Isolated strains were first treated with a serial dilution of antibiotics (a combination of Penicillin G sodium salt and Streptomycin sulfate) (Bioshop Canada) for an exposure period of 24 h. Algal colonies were selected from the lowest antibiotic concentration that had no bacterial growth. Aliquots of the exposed strains were transferred to fresh media, with no antibiotics, after 24 h, followed by pour plate preparation, 1% peptone and 1.5% agar, to evaluate bacterial contamination in addition to microscopic observation. Strains deemed to be axenic were monitored by this method once a month to ensure axenic status was sustained. Strains that were found to be non-axenic were re-treated, but a few non-axenic strains were classified as such if bacterial contamination could not be removed. Only axenic strains were assessed for fatty acid analyses.

DNA extraction, amplification, and sequencing

Genomic DNA of isolated strains was extracted using the commercial kit UltraClean Plant DNA Isolation kit (MoBIO, CA, USA). Cell lysis was achieved by chemical, SDS and mechanical means using a Vortex Genie with horizontal microtube attachment (Scientific Industries, NY, USA). Bead tubes provided by the extraction kit were subjected to three, 5-min bead beating sessions at 3,200 rpm. Samples rested in the tubes on ice for 5 min between bead beating sessions. Upon completion of cell disruption, 20 μL of RNase A (Geno Technology, MO, USA) was added and vortexed gently for 10 min. Extracted DNA was quantified at 260/280 nm by UV-Spectrophotometry (Gensys 10S UV/VIS) and frozen at −20 °C or kept at 4 °C until PCR amplification.

The 18S rRNA forward primer NS1 (White et al., 1990) and reverse primer ITS2 (White et al., 1990) were used to prime PCR at a concentration of 1.0 μmol. Reactions were carried out using Illustra Ready-To-Go PCR beads (GE Healthcare, USA) in a final volume of 25 μL. The PCR reaction was performed using the T100 Thermal cycler (BIORAD, Canada) with the following cycle parameters: initial denaturation 96 °C for 3 min, annealing at 51 °C/1 min, and extension at 72 °C for 1.30 min. This was followed by 29 cycles of denaturation at 93 °C/1.30 min, annealing at 51 °C for 1.30 and extension at 72 °C for 1.30 min, followed by a final extension step at 72 °C for 5 min. Strains that would not amplify by this method were amplified using the forward NS1 primer and reverse 18L primer (Hamby & Zimmer, 1991), along with a touchdown PCR cycle with annealing beginning at 61 °C and reducing by 2 °C every 2 cycles until 51 °C was reached. Gel purification of PCR bands was carried out using the Wizard SV Gel and PCR Clean-Up System (Promega, USA). Modifications were made to the general protocol with the addition of an extra rinse of the column with membrane wash-solution as well as an additional 1-min spin at 16,000 × g with the lid off to aid in evaporation of wash solution. Extracted bands were sent to the Génome Québec Innovation Centre, McGill University, Canada for Sanger sequencing.

Sequencing results were locally aligned using the NCBI Nucleotide BLAST and closest matches assigned based on sequences from the NCBI database. BLAST matches that aligned closely, between 90–99% with culture strains, were selected and added to sequence comparisons in addition to more distantly related species within each family. Only NS1 region sequences greater than 400 base pairs (bp) were included in further phylogenetic analyses. These sequences were deposited into the NCBI database and accession numbers for each sequence were acquired.

Phylogenetic tree reconstruction was performed on the Phylogeny.fr platform (Dereeper et al., 2008; Dereeper et al., 2010), which comprised a series of steps in the analysis. Sequences were aligned and configured with the highest accuracy using MUSCLE (v3.8.31) (Edgar, 2004). After alignment, ambiguous regions containing gaps or poorly aligned regions were removed with Gblocks (v0.91b) using the following parameters: minimum length of a block after gap cleaning = 10; no gap positions were allowed; all segments with contiguous non-conserved positions bigger than eight were rejected; and minimum number of sequences for flank position was 85% (Castresana, 2000). The phylogenetic tree was reconstructed using the neighbour-joining method (Saitou & Nei, 1987; Gascuel, 1997) implemented in the BioNJ program with 1,000 bootstrap replicates (Felsenstein, 1985; Elias & Lagergren, 2007). The analysis involved 45 nucleotide sequences with an average sequence length of 735 bp. Graphical representation and editing of the phylogenetic tree was performed with TreeDyn (v198.3) (Chevenet et al., 2006).

Fatty acid methyl esters (FAME) analysis

To avoid possible confounding effects of bacteria, only axenic isolates (thirty-four in total) were subjected to fatty acid analysis. Isolated strains were grown to stationary phase under previously described growth conditions and 10-mL subsamples were transferred to 14-mL glass tubes with lids and centrifuged at 16,000 rpm (Sorval ST16, Thermo Scientific, MA, USA) for five min. Pelleted biomass was frozen at −20 °C for 48 h followed by 24–48 h of lyopholization with a modulyoD freeze dryer (Thermo Scientific, USA). A direct method of transesterfication of algal fatty acids was performed as per O’Fallon et al. (2007). A 100 ppm nonadecanoic acid methyl ester C19:0 (Sigma Aldrich, Canada) internal standard was added to each sample followed by analysis by Gas Chromatography/Mass Spectrometry (GC/MS).

Methylated fatty acids in hexanes were analyzed on a Varian 450 gas chromatographer with HP-5ms Ultra Inert GC Column, 30 m × 0.25 mm × 0.25 μm (Agilent J&W). A split ratio of 10:1 was used for 3.5 min with a flow rate of 1.0 mL/min, helium carrier gas. The oven temperature was held at 135 °C for 4 min and increased by 4 °C/min until 250 °C, where it was held for 10 min until run-end. A Varian 240 Ion Trap Mass Spectrometer detector was used and fame peaks were identified based upon spectral comparison and retention times to FAME standards: BAME and Supleco C8:0–C24:0 (Sigma Aldrich, Canada). A negative control was included in the FAME process as a quality measure and analyzed under identical sample conditions to ensure peaks obtained were attributable to microalgae samples.

To assess the biofuel potential of the lipid content measured for each isolate, several measures of biodiesel quality were determined using the methodologies and equations outlined in Ramos et al. (2009) and Nascimento et al. (2013). ASTM Standard D675 for biodiesel quality requires a minimum Cetane Number (CN) of 47 and a maximum Iodine Value (IV) of 120.

Copper toxicity bioassays

Five isolates representing diverse taxonomic genera were further characterized to assess their inherent tolerance to copper, a particularly toxic metal to microalgae and a metal that is commonly found in industrial and municipal wastewaters. Genera selected include: Scenedesmus, Monoraphidium, Chlorella, Selenastrum and Microcystis. The Canadian Phycological Culture Collection strain Chlorella kesslerii CPCC266 was also tested as it is a commonly used bioassay reference-strain. A standard 72-h static growth response bioassay was used to assess inherent tolerance to copper. The assay methodology was based on the protocol established by Environment Canada (2007). Prior to commencing the bioassays, all glassware, tips, and applicable materials were acid-washed in a 10% HCL solution for a period of 24 h followed by triplicate rinse with Milli-Q™ filtered water. A stock Copper (II) Sulfate Pentahydrate (32 mg L−1 Cu) solution was prepared with a solution of CHU10 media and sterilized by autoclave. The stock solution was used to aseptically prepare copper concentrations of 0.01, 0.1, 0.32, 1.0 and 3.2 parts per million (ppm) Cu, diluted with sterile CHU10 media. All concentrations and control cultures were prepared in triplicate to a final volume of 25-mL in 50-mL Erlenmyer flasks. Innocula (1-mL) from exponential phase cultures were added to three replicate flasks with each test concentration and maintained under the same growth conditions in a controlled environment incubater with shaker table (Algaetron AG130-ECO) at 22 °C+/− 2 °C under 12 h light/dark cycles at 150 mmol photons m−2 s−1 and 200 rpm. The pH of each replicate was monitored daily using an Oakton pH/Ion 510 bench meter. Fluorescence measurements of chlorophyll a were taken at 24-h intervals using an Aquafluor Hand Held fluorometer, whereby the chlorophyll a solid secondary standard (Turner Designs, MA, USA) was used as a qualitative calibration tool for chlorophyll a measurement. The growth rate was calculated as previously described and data were normalized to percent-control. Growth-rate data were also used to calculate percent inhibition: %I=C−XC×100

where C = average control growth-rate, X = average test-concentration growth-rate.

Results

In total, forty-three algal strains were isolated from all habitat types. Tentative taxonomic assignments were given to all isolates and, with the exception of one cyanobacterial strain (Microcystis sp. Sp21.01), NCBI BLAST searches of partial 18S rRNA sequences were performed (Table 1). All eukaryotic isolates are members of the algal division Chlorophyta, which is a diverse group of green microalgae. Taxonomic assignment and BLAST matches were fairly congruent, but there were several exceptions. All isolates were initially characterized for growth-rate under standardized conditions (Table 1), and for all strains, exponential growth was most commonly observed to last up to three days. Fastest growing strains came from stormwater ponds and included Scenedesmus sp. Sp19.011 and Desmodesmus sp. Sp19.15, which doubled their biomass in just over 13 h. In contrast, the slowest growing strains Scenedesmus sp. Sp21.12 and Chlorella sp. Sp21.20 had generation times between 40–42 h. Over 60% of the strains were found to have a generation time less than 24 h (Table 1).

Table 1 Summary list of isolated strains and their associated isolation medium, source location, tentative taxonomic assignment and 18S rRNA sequence similarity with BLAST taxa.

Growth rate and generation time under standardized growth conditions over seven-days are also included. With the exception of one cyanobacterial strain (Microcystis sp.) which has been listed because of its inclusion in the bioassay results, all eukaryotic strains were subjected to 18S rRNA sequencing using NS1 and ITS2/18L primers. See Vincent & Kirkwood (2014) for more information on stormwater and reference pond characteristics.

Strain ID	Isolating medium	Source location	Taxonomic assignment	Closest BLAST match	Similarity (%)	Sequence size (bp)	Growth rate (d−1)	Generation time (h)	
Sp1.41	CHU10	SWP 1	Scenedesmus sp.	Desmodesmus armatus KF673362	99	1,160	1.14	14.6	
Sp1.43	CHU10	SWP 1	Desmodesmus sp.	Desmodesmus intermedius FR865701	98	1,253	0.74	21.1	
Sp1.44	CHU10	SWP 1	Chlorella sp.	Chlorella sp. KP262476	99	1,580	0.75	22.2	
Sp1.46	CHU10	SWP 1	Dictyosophaerium sp.	Dictyosphaerium ehrenbergianum GQ487213	99	474	0.69	24.0	
Sp1.50	CHU10	SWP 1	Ankistrodesmus sp.	Ankistrodesmus gracilis AB917098	98	1,618	1.15	14.5	
Sp1.52	CHU10	SWP 1	Chlorella sp.	Mychonastes rotundus GQ477053	83	343	0.97	17.2	
Sp11.30	CHU10	SWP 11	Scenedesmus sp.	Scenedesmus obliquus FN298925	99	911	0.51	32.6	
Sp12.07	BG11	SWP 12	Chlorella sp.	Coelastrum astroideum GQ375093	99	1,518	0.55	30.2	
Sp12.21	CHU10	SWP 12	Chlorella sp.	Coelastrum astroideum GQ375093	100	850	1.04	16.1	
Sp12.36	CHU10	SWP 12	Scenedesmus sp.	Acutodesmus obliquus AB917101	99	1,614	0.90	18.6	
Sp13.17	CHU10	SWP 13	Chlorella sp.	Mychonastes huancayensi GQ477050	100	1,445	0.99	17.7	
Sp14.35	CHU10	SWP 14	Selenastrum sp.	NS	–	–	0.96	17.41	
Sp16.26	CHU10	SWP 16	Scenedesumus sp.	Scenedesmus sp. FN298925	99	1,431	0.90	18.6	
Sp16.34	CHU10	SWP 16	Scenedesmus sp.	Scenedesmus obliquus KJ676128	99	714	1.10	15.1	
SpU.9*	CHU10	SWP 16	Chlamydomonas sp.	Chlamydomonas inflexa FR865584	97	1,143	0.59	28.1	
Sp17.013	BG11	SWP 17	Monoraphidium sp.	Monoraphidium convolutum HM483515	99	1,309	0.92	18.1	
Sp17.022	BG11	SWP 17	Chlorococcum sp.	Chlorococcum ellipsoideum U70586	95	423	0.65	25.6	
Sp17.25	CHU10	SWP 17	Chlorella sp.	Chlorella luteoviridis FR865678	99	1,482	0.98	17.0	
Sp17.38	CHU10	SWP 17	Monoraphidium sp.	Monoraphiridium sp. JN187941	98	873	1.12	14.9	
Sp19.010	BG11	SWP 19	Desmodesmus sp.	Desmodesmus communis KF864475	99	1,178	0.64	26.0	
Sp19.011	BG11	SWP 19	Scenedesmus sp.	Scenedesmus acuminatus AB037088	99	1,073	1.24	13.5	
Sp19.015	CHU10	SWP 19	Monoraphidium sp.	Monoraphiridium contortum AY846382	97	588	0.72	23.1	
Sp19.15	CHU10	SWP 19	Desmodesmus sp.	Desmodesmus pannonicus FR865712	98	821	1.22	13.7	
Sp19.40	CHU10	SWP 19	Chlamydomonas sp.	Chlamydomonas debaryana JN903975	99	651	1.10	15.1	
Sp21.01	BG11	SWP 21	Microcystis sp.	–	–	–	0.69	24.0	
Sp21.02	BG11	SWP 21	Chlorella sp.	Chlorella luteoviridis FR865678	99	718	0.80	20.7	
Sp21.12	CHU10	SWP 21	Scenedesmus sp.	Scenedesmus sp. FR865732	99	1,483	0.40	41.8	
Sp21.14	CHU10	SWP 21	Chlorella sp.	Chlorella luteoviridis FR865678	99	449	0.75	22.1	
Sp21.20	CHU10	SWP 21	Chlorella sp.	Chlorella luteoviridis FR865678	100	1,336	0.42	39.8	
Sp21.37	CHU10	SWP 21	Ankistrodesmus sp.	Monoraphidium contortum KM067465	94	626	1.10	15.1	
Sp21.23	CHU10	SWP 21	Desmodesmus sp.	Desmodesmus intermedius FR865701	99	1,245	1.07	15.6	
Sp23.13	CHU10	RP 23	Chlorella sp.	Micractinium inermum KF597304	100	1,005	0.55	30.3	
Sp24.1	CHU10	RP 24	Scenedesmus sp.	Scenedesmus sp. FN298925	100	257	0.55	30.3	
Sp24.05	BG11	RP 24	Desmodesmus sp.	Desmodesmus pannonicus FR865712	99	1,158	0.50	33.2	
LO47	CHU10	Lake ON	Ankistrodesmus sp.	Monoraphidium griffthii AY846383	100	1,454	0.74	22.5	
LO48	CHU10	Lake ON	Scenedesmus sp.	Scenedesmus obliquus FR865738	100	1,119	1.00	16.7	
LO49	CHU10	Lake ON	Chlorella sp.	Micractinium sp. AB9187105	100	350	1.08	22.2	
LO51	CHU10	Lake ON	Chlorella sp.	Chlorella luteoviridis FR865678	100	585	0.58	29.1	
WW3	CHU10	WWTP	Chlorella sp.	Coelastrum microporum JQ315528	98	497	0.63	25.3	
WW5	CHU10	WWTP	Scenedesmus sp.	Acutodesmus obliquus KF144164	99	597	0.58	28.7	
WW8	CHU10	WWTP	Chlorella sp.	Chlorella luteoviridis FR865678	99	761	0.80	20.7	
WW27	CHU10	WWTP	Chlorella sp.	Chlorella luteoviridis FR865678	99	1,304	0.77	21.7	
WW39	CHU10	WWTP	Desmodesmus sp.	Desmodesmus intermedius KF673371	99	1,572	0.66	25.4	
Notes:

* The source location for strain SpU.9 is either SWP 16 or SWP 17.

NS, Strain not successfully sequenced; WWTP, Wastewater Treatment Plant; SWP, Stormwater Pond; RP, Reference Pond; Lake ON, Lake Ontario.

A subset of twenty-seven isolates with NS1sequences greater than 400 bp were used with eighteen NCBI database-taxa to reconstruct an unrooted neighbour-joining phylogenetic tree (Fig. 1). All strains included in the tree belong to the Chlorophyceae and Trebouxiophyceae classes. Tree topology distinctly groups isolates into families and genera. In total, the tree demarcates study-isolates into seventeen unique phylotypes based on branch lengths ≤ 0.001 base-pair substitutions per site. Most isolates had sequence matches with taxa from the BLAST database, with the exception of only a few including: Monoraphidium sp. Sp19.015, Scenedesmus sp. Sp19.011, Desmodesmus sp. Sp19.15 and Chlamydomonas sp. SpU9.

Figure 1 Unrooted Neighbor-joining tree for 18S rRNA NS1-region sequences greater than 400 bp for algal isolates and related taxa from the NCBI database.

Bootstrap scores are based on 1,000 replicates. NCBI database accession-codes have been included with strain names.

FAME results for thirty-four axenic isolates identified seven different fatty acids, including saturated (lauric acid (C12:0) and palmitic acid (C16:0)), monounsaturated (oleic acid (C18:1)) and polyunsaturated (linoleic acid (C18:2), hexadecatetraenoate (C16:4(n-3) and eicsoatetraenoic acid (C20:4)) forms. Palmitic and oleic fatty acids were the predominant lipid components found in all strains. Several strains including Chlamydomonas sp. SpU9, Chlamydomonas sp. Sp19.40, Ankistrodesmus sp. LO47 and Chlorella sp. LO51 had a large percentage (70–90%) of their total FAME lipid as palmitic acid. Desmodesmus sp. WW39 and Scenedesmus sp. Sp16.34 had oleic acid greater than 80% of their total FAME lipid profile. Chlorococcum sp. Sp17.022 was the only strain observed to have oleic acid C18:1 present in both cis- and trans-forms. One of the most unique profiles belongs to Chlorella sp. Sp21.02, which was found to have 53% of its lipids in polyunsaturated forms (Fig. 2). There was no apparent distinction of FAME profiles among strains isolated from different source-locations (see Supplemental Information).

Figure 2 FAME profiles for thirty-four axenic algal isolates based on the relative abundance of fatty acids detected by GC-MS.

To determine if there was any congruence among strains with respect to lipid composition and phylogenetic relatedness, the FAME profiles of strains sharing a phylotype with at least one other strain were compared using hierarchical cluster analysis (Fig. 3). The Bray-Curtis similarity dendrogram grouped strains of the same phylotype as being most similar based on their FAME profile. The cophenetic correlation coefficient of 0.87 indicates that the dendrogram adequately preserved the pairwise distances among strains. Bootstrap scores were mixed, but were moderate to strong for most nodes in the dendrogram. Strains Scenedesmus sp. Sp12.36 and Scenedesmus sp. LO48 are the same phylotype and had the most similar FAME profiles (> 95% similarity). In contrast, Ankistrodesmus sp. LO47 and Ankistrodesmus sp. Sp1.50, which are the same phylotype, were only about 55% similar based on their FAME profiles. Even so, they were grouped as most similar among isolates in the cluster dendrogram (Fig. 3).

Figure 3 Comparison of FAME profiles among distinct phylotypes with two or more isolated strains using Bray-Curtis Unweighted Pair Group Method with Arithmetic Mean (UPGMA) cluster analysis.

Strains of the same phylotype appear as the same colour in the dendrogram.

With respect to biodiesel quality, twenty-eight isolates had a CN greater than 47, and thirty isolates did not exceed an IV of 120 (Table 2). In combination, twenty-eight isolates had acceptable CN and IV values according to the ASTM standard. The degree of unsaturation among isolates was quite varied, ranging from very low (10.6) for Chlorella sp. Sp23.13 to very high (120) for Ankistrodesmus sp. Sp1.50. Long-chain saturation factor also varied among isolates, where isolate Scenedesmus sp. Sp16.34 had the lowest value (0.55) by an order of magnitude, and Chlorella sp. Sp23.13 had the highest value (8.94). The Saponification Value was less varied among strains, where most had values varying from 200–210. Strain Chlamydomonas sp. SpU9 had a notably high SV of 241. The majority of isolates had negative-integer values for Cold Filter Plugging Point (CFPP). Scenedesmus sp. Sp16.34 in particular, had a CFPP as low as −15 °C. In contrast, Sp23.13 had a relatively high CFPP at 12 °C.

Table 2 Biodiesel properties of algal isolates based on their fatty acid profiles.

The ASTM Biodiesel Standard D675 requirement for CN is a minimum value = 47 and IV maximum = 120. Degree of Unsaturation is a weighted sum of the masses of monounsaturated and polyunsaturated fatty acids, and Long-Chain Saturation Factor is a weighted sum of long-chain fatty acids (C16, C18, C20, C22 and C24). Saponification value is equivalent to milligrams of potassium hydroxide required to saponify 1 g of oil.

Strain ID	Cetane number	Iodine value	Degree of unsaturation (wt%)	Long-Chain saturation factor (wt%)	Saponification value (mg)	Cold filter plugging point (°C)	
Sp1.41	59.4	51.9	57.7	4.44	207	−3	
Sp1.43	61.0	45.0	50	5.12	209	0	
Sp1.44	53.7	74.3	66.9	4.9	207	−1	
Sp1.46	58.0	56.3	62.5	4.01	210	−4	
Sp1.50	18.9	207	120	2.37	215	−9	
Sp1.52	55.4	67.0	60.4	4.51	209	−2	
Sp12.07	44.6	108	76.7	4.11	211	−4	
Sp12.21	57.4	59.7	56.6	4.1	207	−4	
Sp12.36	55.3	68.7	68.6	3.45	206	−6	
Sp14.35	60.3	48.3	53.7	4.69	208	−2	
Sp16.26	59.1	53.7	59.8	4.02	206	−4	
Sp16.34	45.6	110	104	0.55	200	−15	
SpU9	65.6	13.3	37.3	8.26	241	9	
Sp17.013	36.3	139	85.7	4.16	214	−3	
Sp17.022	66.0	22.7	46.1	6.44	214	4	
Sp17.25	61.3	42.9	47.6	6.02	210	2	
Sp19.010	55.0	68.0	60.5	4.4	210	−3	
Sp19.011	54.7	68.0	57.7	4.59	212	−2	
Sp19.15	56.4	63.2	59.8	4.43	208	−3	
Sp19.40	68.8	10.7	11.9	8.81	216	11	
Sp21.02	25.3	186	118	3.51	206	−5	
Sp21.12	58.1	56.6	56.6	4.71	208	−2	
Sp21.14	55.0	68.5	62	5.15	208	0	
Sp21.20	53.9	71.9	60	5.76	210	2	
Sp21.37	60.7	46.5	51.7	4.83	208	−1	
Sp23.13	69.1	9.60	10.6	8.94	216	12	
Sp24.05	47.7	94.8	69	4.18	213	−3	
LO47	56.1	61.1	42	7.24	215	6	
LO48	53.9	73.9	71.3	3.44	206	−6	
LO49	52.6	78.6	66.2	5.15	207	0	
LO51	67.8	14.5	48.8	6.88	217	5	
WW3	53.9	73.5	73.1	2.67	207	−8	
WW39	49.5	93.0	90.6	1.45	203	−12	
WW5	31.0	162	35.2	2.01	210	−10	

To assess the inherent tolerance of algal isolates to the common wastewater contaminant copper, five representative-strains from a variety of taxonomic groups were subjected to static growth-response bioassays. The cyanobacterial strain Microcystis sp. Sp21.01 was the least tolerant to copper, having complete growth inhibition at 0.32 mg·L−1 or higher. The two stormwater pond isolates Monoraphidium sp. Sp17.38 and Selenastrum sp. Sp14.35 had significantly higher (Student’s t-test p < 0.05) tolerance to 0.32 mg·L−1 copper than the reference strain Chlorella kesslerii CPCC26. The most tolerant strain was Selenastrum sp. Sp14.35, which had the highest growth rate at 1 mg·L−1 copper. No strains could grow at the highest test concentration 3.2 mg·L−1. At municipal-wastewater relevant concentrations of copper, all test strains, with the exception of Microcystis sp. Sp21.01, had minimal to no growth inhibition at the highest copper concentration 0.1 mg·L−1 (Table 3).

Table 3 Comparison of algal tolerance to the range of copper concentrations typically found in municipal-wastewater systems (Environment Canada, 2001).

Copper concentrations previously measured for source stormwater-ponds are also included (Vincent & Kirkwood, 2014).

		Copper (% inhibition)	
Strain ID	Source location copper (mg·L−1)	0.01 mg·L−1	0.1 mg·L−1	
Chlorella kesslerii CPCC266	N/A	0	0	
Chlorella sp. Sp21.20	0.0021	5.25	5.84	
Scenedesmus sp. Sp11.30	0.0035	0	0	
Selenatrum sp. Sp14.35	0.0061	0	1.04	
Monoraphidium sp. Sp17.38	0.0040	0	0.22	
Microcystis sp. Sp21.01	0.0021	0.86	53.4	
Note:

Average percent-inhibition is reported for each test concentration: N/A, information not available.

Discussion

Diversity of algal isolates

All isolated algal strains (with the exception of one cyanobacterial isolate) belong to the Chlorophyta, a class of algae that includes taxa commonly used in algal biofuel research and application (Mata, Martins & Caetano, 2010). In particular, Chlorella sp. and Scenedesmus sp. are the most commonly used for biofuel feedstock, and our study isolated several strains from each of these genera. Chlorophyte taxa such as the strains from our study are easy to culture, which in part, is likely why they are common in biofuel studies. Yet, these taxa have also been shown to be relatively good lipid producers (Mata, Martins & Caetano, 2010), which increases their utility as feedstock in biodiesel production. As such, all source locations in this study, and in particular stormwater ponds and the municipal wastewater treatment plant, were fruitful habitats for isolation of lipid-rich strains suitable for biodiesel production.

Several discrepancies between taxonomic assignment and BLAST match (Table 1) was not entirely unexpected as this can be a common scenario. Though many chlorophyte species exhibit type-features, such as spine and coenobium formation, culture condition will often affect these characteristics used in identification. Morphological heterogeneity also makes algal identification challenging, so it is possible that some strains were misidentified. In contrast, 18S rRNA sequencing can be useful in distinguishing phylotypes of morphologically similar strains (Friedl, 1995; Lewis & McCourt, 2004). This was evident in the phylogenetic tree reconstruction, which provided a clear resolution to generic assignment for isolates in the tree. In most instances, morphological similarities of clade members were supported by genetic homology presented in the tree organization. BLAST sequence searches for some of the more closely related Scenedesmus, Desmodesmus and Coleastrum taxa often returned similar local alignment matches for query cover and percent identity, making them difficult to confirm solely based on closest-match results. Again, the phylogenetic tree topology provided useful clarification of misidentifications found in the BLAST database. Sequence databases are known to have a small percentage (∼20%) of their sequence collection to be mislabelled or misidentified (Bridge et al., 2003, Vilgalys, 2003), and algal systematics is currently in flux. As such, definitive taxonomic and phylogenetic assignment for strains in this study will likely require further classification in the future.

Algal growth and fatty acid production

Rapid growth rates are an important characteristic of biofuel feedstock, since fast-growing strains would have increased yields in a shorter time-frame. According to Griffiths & Harrison (2009), the average doubling time for green microalgae is 24 h, which corresponds to a growth-rate (μ) = 0.69 d−1. Of the forty-three isolates evaluated, the majority of strains were able to double their biomass in 24-h or less (Table 1). Twelve isolates from our study could grow ≥ 1.0 d−1 (Table 1), which is considered to be a relatively fast growth-rate for microalgae in general, and thus a desirable trait for biofuel-feedstock. Many of the fastest growing taxa were from stormwater ponds and included closely related genera such as Scenedesmus, Desmodesmus, and Ankistrodesmus (Fig. 1). If these phylogenetically related taxa were to be targeted for isolation from other systems, our results from a subset of strains would suggest that they may have similar fatty acid profiles (Fig. 2), even for those of the same phylotype (Fig. 3). However, calculations presented in Table 2 show phylogenetically similar strains (e.g., Sp12.07 and Sp12.21) have differing biodiesel qualities. This finding concurs with the Culture Collection of Algae at Goettingen University (SAG) study that found similarities among fatty acid distribution patterns were only found at phyla and class levels (Lang et al., 2011). When accounting for all strains that were assessed for their fatty acid profiles (Fig. 2; Table 2), it is apparent that the diverse array of taxa, even from the same class, exhibited a broad range of variation in fatty acid composition. The observed variation in growth rate and fatty acid profile among strains in our collection supports the continued need to characterize microalgae at the strain-level, rather than targeting a certain family or genera, when bioprospecting for biofuel candidates.

Common fatty-acid esters found in manufactured biodiesel include: palmitic C16:0 (hexadecanoic) acid, stearic C18:0 (octadecanoic) acid, oleic C18:1 (9(Z)-octadecenoic) acid, linoleic C18:2 (9(Z),12(Z)-octadecadienoic) acid, and linolenic C18:3 (9(Z),12(Z),15(Z)-octadecatrienoic) acid (Knothe, 2008). The dominant saturated fatty-acid in most strains from this study was C16:0 palmitic acid, and in many cases, represented approximately 50% of the total fatty-acid profile. The dominant unsaturated fatty acid in most strains was C18:1 oleic acid, and it too represented about 50% of the total fatty acid profile among several strains. Biofuels with high oleic acid content have been reported to have reasonable ignition quality, combustion heat, CFPP, oxidative stability, viscosity, and lubricity (Chen et al., 2012). Hu et al. (2008) confirm that fatty acids with carbon chain lengths from 16–18 units are ideal precursors for biodiesel production. Not only does carbon chain length provide a basis for deriving biodiesel, but specifically, the saturation versus unsaturation garners characteristics that make the fuel more versatile.

Potential for biodiesel production

Biodiesel derived from microalgae with more saturation provide a higher CN, would have lower NOx emissions, and have shorter ignition delay time (Cherisilp & Torpee, 2012). Nevertheless, this can come at a cost since lower temperatures would cause saturated fatty-acids to solidify due to their high melting point (Doğan & Temur, 2013). Unsaturated fatty-acids require less heating and are often liquids at room temperature, however, the higher the number of double-bonds, the more prone the fuel is to producing NOx emissions (Gopinath, Puhan & Nagarajan, 2010). Gopinath, Puhan & Nagarajan (2010) claim that having a 50/50 blend of saturated and unsaturated fatty acids produces better thermal efficiency and reduces NOx emissions. Based on this biodiesel characteristic, many of the strains isolated in this study have desirable fatty-acid profiles reflecting a ∼50/50 saturated/unsaturated blend.

TAG commonly accumulate in many algal species as a storage product. As culture resources are depleted and algal growth slows, so does the production of new membrane compounds such as phosphoglycerides, glycosylglycerides and sterols. Under these conditions, fatty-acid production is diverted to TAG synthesis (Guischina & Harwood, 2006). This metabolic shift results in a trade-off impacting biomass production. Griffiths & Harrison (2009) reported that cultivation conditions focusing on biomass productivity instead of lipid production per cell may ultimately be more beneficial and more efficient in increasing total lipid productivity. Several fast growing isolates from this study (including Scenedesmus sp. Sp1.41, Scenedesmus sp. Sp19.011, and Desmodesmus sp. Sp19.15) have ideal biodiesel properties based on their CN, IVs and Degree of Unsaturation (Table 2). Hence, these strains may represent the best candidates for optimal lipid yields based on growth-rate alone.

Algal copper tolerance

Although a select, but diverse group, of isolates were tested for copper tolerance, all chlorophyte strains exhibited high tolerance at wastewater-relevant concentrations (Table 3). Thus, it is plausible that many chlorophyte isolates in our collection would likely have some inherent tolerance to copper if one extrapolates from the copper bioassay results. Others have also recognized the higher tolerance thresholds of chlorophyte microalgae to wastewater conditions, focusing primarily on Chlorella and Scenedesmus strains (Lau, Tam & Wong, 1995; Bhatnagar et al., 2010; Ruiz-Marin, Mendoza-Espinosa & Stephenson, 2010). Interestingly, the only non-chlorophyte strain in our study (Microcystis) was uniquely less tolerant to copper than the chlorophyte strains, even though it was also isolated from a stormwater pond. It is also interesting that two of the stormwater pond isolates Monoraphidium sp. Sp17.38 and Selenastrum sp. Sp14.35 were notably tolerant to copper at concentrations orders of magnitude higher than their source stormwater ponds (Table 3). Therefore, these strains in particular may be ideal candidates for industrial-wastewater feedstock since they are relatively fast growers (Table 1), have ideal fatty-acid profiles (Fig. 2; Table 2), and have inherent tolerance to the algicidal-metal copper (Table 3).

Conclusions

Bioprospecting of algal strains for use in biofuel production is a relatively new trend. Though our study is the first to target biofuel bioprospecting efforts in stormwater ponds and a municipal wastewater system, the dominant taxa isolated, including Chlorella, Scenedesmus, and Desmodesmus, are cosmopolitan in freshwater environments. Strains identified as the species Scenedesmus obliquuss and Chlorella leuteoiviridis in blast matches were prevalent in both stormwater ponds and the wastewater treatment facility, and they typically out-competed other species during cultivation. Zhou et al. (2011) found similar results in isolations of microalgae from wastewater sites in the United States, isolating several species of Chlorella. Past efforts to isolate microalgae from various habitats including wastewater systems was mainly for wastewater remediation. Chlorella and Scenedesmus are common taxa isolated for this purpose, especially for nutrient removal (Tam & Wong, 1989; Tam & Wong, 1990; González, Cañizares & Baena, 1997; Woertz et al., 2009; Chinnasamy et al., 2010). Nutrient removal by microalgae cultivated in wastewater would provide a value-added benefit during biofuel-feedstock production.

While the results presented here are only a first step in assessing algal strains from wastewater habitats as biofuel-feedstock candidates, the ultimate goal is to establish and select strains most tolerant to municipal and/or industrial wastewater conditions. The use of wastewater to support algal biomass production is not a new approach, but it is for biofuel feedstock production. Several studies have all concluded that there is great potential for algal biofuel production using wastewater (Pittman, Dean & Osundeko, 2011; Bhatnagar et al., 2011; Christenson & Sims, 2011). Yang et al. (2011) looked at the total water and nutrient requirements of microalgae-based biofuels and found that biofuels sourced from wastewater decreased total water requirements by 90% in comparison to freshwater sourcing without recycling. What has not been investigated explicitly is the viability of using wastewater on a continuous basis in a pond or photobioreactor system growing single or mixed strains of microalgae. Broad variations can occur in wastewater quality, in addition to fluctuations in toxic-metal constituents (Coale & Flegal, 1989; Ahluwalia & Goyal, 2007). In the case of municipal wastewater, the nutrient and contaminant content can vary daily depending on the source-population. As such, future studies of biofuel microalgae, including promising isolates from this study, would need to be assessed for their tolerance and resilience to fluctuating wastewater conditions while maintaining an acceptable growth rate and lipid yield.

Supplemental Information

Supplemental Information 1 Growth raw data.

Click here for additional data file.

Supplemental Information 2 FAME data.

Click here for additional data file.

Supplemental Information 3 Copper assay data.

Click here for additional data file.

Supplemental Information 4 Principal component analyses of fatty acid profiles.

Principal component analyses of fatty acid profiles for thirty-four algal strains labelled by source location: Black circle = Stormwater Pond, Orange square = natural reference site, Red Diamond = Municipal Wastewater Treatment Plant. Vector lines represent individual fatty acids.

Click here for additional data file.

Additional Information and Declarations

Competing Interests

Author Contributions

DNA Deposition

Data Deposition

The authors declare they have no competing interests.

Rebecca Massimi conceived and designed the experiments, performed the experiments, analyzed the data, wrote the paper, prepared figures and/or tables, reviewed drafts of the paper.

Andrea E. Kirkwood conceived and designed the experiments, analyzed the data, contributed reagents/materials/analysis tools, wrote the paper, prepared figures and/or tables, reviewed drafts of the paper.

The following information was supplied regarding the deposition of DNA sequences:

NCBI accession numbers: KU517410; KU517414; KU517413; KU517412; KU517411; KU517415; KU517416; KU517417; KU517436; KU517418; KU517419; KU517420; KU517421; KU517423; KU517422; KU517428; KU517424; KU517425; KU517434; KU517431; KU517430; KU517435; KU517432; KU517433; KU517429; KU517427; KU517426.

The following information was supplied regarding data availability:

The raw data has been supplied as Supplemental Dataset Files.

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
