# Peer review of "Screening microalgae isolated from urban storm- and wastewater systems as feedstock for biofuel"

_PeerJ, doi:10.7717/peerj.2396_

## Round 0.1 · original submission · Major Revisions

The manuscript is relevant. Once the corrections indicated are made, it should be accepted. The recommendations of the reviewers are quite accurate and directly affect the aspects to be improved.

Reviewer 1 ·

Basic reporting

In this ms the authors isolated several microalgal strains from storm- and wastewater ponds, characterized them taxonomically by gene sequencing, measured their growth rates and produced FAME profiles. They found many fast growing strains with FAME profiles suitable for biodiesel production. The text is clearly written although some statements lack reference. The works done is of good quality and scientifically sound and the ms should be published after revision. For clarity, the diiscussion section should be divided in items e.g.
-Diversity of algal isolates
-Growth characteristics and tolerance to copper
-Potential for biodiesel production

Experimental design

The research question of the ms is simple and well defined. However, I have some comments on the approach used for establishing the microalgal cultures and some calculations of cell concentrations.

1. The antibiotics treatment approach to reduce bacteria loads is a valid one for the reasons given by the authors (it reduces interference of bacteria in the FAME profile), although bacteria contamination is normally a minor problem during the exponential growth phase of algal cultures. I wonder how much this antibiotics treatmente influenced growth rates and FAME profiles of the strains used in the experiments. If some of the strains were so susceptible to the antibiotics cocktail that they were removed from the experiments, other strains could have their growth rates negatively affected as well. Also, large-scale production of microalgal biomass will not be axenic thus the FAME profile is likely to change under this conditions. Antibiotics susceptibility could be the reason for e.g. low growth rates of some strains (like Scenedesmus sp. Sp21.12 and Chlorella sp. Sp21.20) This should be acknowledged in the discussion.

2.Can you assure you achieved clonal algal cultures? The plate spreading technique is a widespread, straightforward way to isolate fast growing algae, but it does not necessarily produce clonal strains (i.e. a culture originated from one cell or coenobium). This should be taken into account since the authors acknowledged (based on references) that strain variability does exist.

Validity of the findings

Experiments are scientifically sound. Results are of good quality and relevant to the field of biofuels. Conclusions are supported by the results. DNA sequences of microalgal strains used in this work should be made available in a public database and accession numbers must be provided as supplementary material.

Additional comments

In the title and throughout the text change algae for microalgae, as the ms deals specifically with these types of microorganisms. Suggestion for the title: Screening microalgae isolated from urban storm- and wastewater systems as feedstock for biofuel. Abstract should me more informative giving e.g. growth rates ranges, maxima, etc. for selected strains that the authors found to be suitable for biofuel feedstock.


line 17: replace "were tolerant of" with "were tolerant to"

line 35 replace could with can

lines 51-54. These statements need references.

line 70 Replace "Strain Collection and Isolation" for "Sample collection and strain isolation".

line 83. Why was the nitrogen concentration of BG11 medium reduced so much? If it was to bring it close to wastewater N concentrations (as stated in the following sentence) then the typical average or range of N concentration in the storm and wastewater ponds sample in the study should be shown.

line 79. Replace grown in for transferred to

line 86. Add reference for f2 medium. Could you please supply nutrient concentration of surveyed ponds as supplementary material? Would be helpful to compare with nutrient concentration of the media used for isolation.

line 131 replace "to prime PCR reactions at a concentration" with "to prime PCR at a concentration"

line 189. species name should not be in capitals. Please check this throughout the text.

line 215. sp. after genus name should not be in italics. Please check this throughout the text.

line 246-248. This belongs to M&M.

line 251. Replace "does a good job of preserving" with preserved.

lines 257-259 This belongs to the M&M section.

lines 259-260 This sentence belongs to the discussion

line 263 Unsaturation should not begin with capitals

line 296 replace " for biofuel strains" with "for isolation of lipid-rich strains suitable for biodiesel production"

lines 301-302 This statement needs a reference.

Figure 3 could be moved to supplementary material. Add species names for the isolates at the tip of the branches.

Table 3 and Figure 5 show basically the same information. I suggest removing Figure 5.

Reviewer 2 ·

Basic reporting

My major concerns with this study are related to the taxonomic part and incongruencies between the phylogenetic analysis and the fatty acid analysis.
1/ The authors have omitted the % similarity from table 1. A similarity >90% is indicated in the legend, but the authors need to be more specific because 10% variation may mean a lot. Different species may have 99% similarity based on the 18S fragment sequence. Two strains having 91-93% similarity could easily belong to different genera…add % similaity to closest match in Table 1.
2/The inconsistency between 18S tree and data in table 2 needs to be explained. For instance, Isolates 12.07 and 12.21 are identical in the 18S tree but differ significantly in the Table 3 parameters. Idem 1.43 and 21.23, 21.20 and 23.13. There are more such cases.
3/I did not find WW8 and WW28 in the Table 2, though they are listed in 18S phylogenetic tree. The authors will need to revise the whole set of isolates and make sure the same set appears in all tables and figures.
Line 288-313 are not convincing; need rewriting focusing on the main findings of the manuscript. Overal discussion needs to be more focused on the results of the present study.

Additional
Monoraphidium sp. Sp17.38 and Selenastrum sp. Sp14.35 are not in the tree (fig 1). Why? Make sure you get the right taxonomic assignment for these two as they are apparently the most interesting for further investigations.

Please Do not use sp. In italic

Experimental design

Authors need to revise their methodology in order to confirm inconsistencies between FAME and 18S analysis.

Validity of the findings

.

Additional comments

.

---

## Round 0.2 · accepted · Accept

Regarding the two issues raised by reviewer #1 (use of antibiotics and character not clonal of crops), the responses of the authors are appropriate.

Regarding the remarks of ref # 2, authors should notice that both relate to the inconsistencies between the 18S and morphology identification. They could also consider that what their FAME analysis results probably demonstrate is nothing more than a normal variation among microalgae.